# Changes in Histological Structure, Interleukin 12, Smooth Muscle Actin and Nitric Oxide Synthase 1. and 3. Expression in the Liver of Running and Non-Running Wistar Rats Supplemented with Bee Pollen or Whey Protein

**DOI:** 10.3390/foods11081131

**Published:** 2022-04-14

**Authors:** Piotr M. Jarosz, Patryk P. Jasielski, Michał K. Zarobkiewicz, Mirosław A. Sławiński, Ewelina Wawryk-Gawda, Barbara Jodłowska-Jędrych

**Affiliations:** Chair and Department of Histology and Embryology with Experimental Cytology Unit, Medical University of Lublin, Radziwiłłowska Street 11, 20-080 Lublin, Poland; patryk.jasielski111@gmail.com (P.P.J.); michal.zarobkiewicz@umlub.pl (M.K.Z.); miroslaw.slawinski92@gmail.com (M.A.S.); ewelina.wawryk-gawda@umlub.pl (E.W.-G.); barbara.jodlowska-jedrych@umlub.pl (B.J.-J.)

**Keywords:** bee pollen, whey protein, liver, interleukin-12, endothelial nitric oxide synthase, smooth muscle actin

## Abstract

**Introduction:** Bee pollen is a natural substance obtained from flowers by bees. It is a rich source of protein, vitamins and minerals. It can be used as a dietary supplement. Bee pollen has been investigated for the treatment of some diseases with promising potential. It can be helpful in supportive therapy for dyslipidemia, metabolic syndrome, diabetes type 2, as well the prevention and control of coronary heart disease and myocardial infarction. Whey protein is a rich source of amino acids. It is a basic dietary supplement for many athletes, both professional and amateur. It stimulates muscle growth and provides nutrition for cachectic patients. **Aim of the study:** The objective of the study was to assess the impact of dietary supplementation of bee pollen or whey protein on the Wistar rat liver histological structure and expression of interleukin 12, smooth muscle actin and nitric oxide synthases among running and non-running rats. **Material and methods:** Thirty male Wistar rats were divided into six equal groups, three running and three non-running. Among both there was one control, one supplemented with bee pollen and one receiving whey proteins. After 8 weeks, all animals were decapitated and their livers were collected. Five micrometer thick slides were prepared and used for classical histological staining and immuno-histochemistry. ImageJ image analysis software was used to measure optical density and immunohistochemistry profile coverage. **Results:** Among all groups, morphology of liver was similar. In the running control group, expression of interleukin-12 (IL-12) was decreased as well as expression of endothelial nitric oxide synthase (eNOS) in a group of bee pollen supplemented rats. No significant changes in α- smooth muscle actin (α-SMA) expression was observed. **Conclusions:** Bee pollen is proving to be a questionable choice for athletes as an alternative to whey protein. Bee pollen supplementation affects hepatocyte cellular activity and has hepatoprotective effects. Whey protein performs worse in this regard. Lower antioxidant properties were found in groups supplemented with bee pollen than with whey protein.

## 1. Introduction

Bee pollen is a flower pollen collected by bees and used as nourishment for the queen bee. A great variety of nutrients have been found in pollen, including carbohydrates, proteins, fats, vitamins, micro-elements, enzymes, essential oils, etc. [1]. Bee pollen is rich in protein (25%) and contains all the essential amino acids, including those exogenous for humans. Moreover, pollen also contains approximately 6% essential and aromatic oils and 51% polyunsaturated fatty acids (PUFAs). Bee pollen contains a variety of vitamins and 11 different carbohydrates, comprising 35–61% of pollen. The carbohydrates are mainly glucose and fructose [2]. Bee pollen contains ca. 5% of different substances such as polyphenolic substances (e.g., flavonoids, organic phenols, ketones) [3,4]. Some floral origins, due to specific compounds, have antifungal, antimicrobial, antiviral, anti-inflammatory and immunostimulant potential [3]. Current studies suggest that bee pollen might help in the treatment of dyslipidemia and the prevention and control of coronary heart disease and myocardial infarction [5,6,7]. It can also protect other organs of the body from free radical damage produced by chemotherapy drugs [8].

Whey protein is a mixture of proteins isolated from whey, the liquid by-product of cheese production [9]. It is considered a dietary supplement commonly used in sport to enhance regeneration and hypertrophy of muscles and is also used for cachectic patients [10].

The data show that athletes place high emphasis on the use of dietary supplements, although their role in supporting training and body building is not exponential. Up to 40–100% of athletes admit to using various supplements, including sports foods, vitamins, probiotics, minerals, protein and creatine [11]. A study by Frączek et al. found that protein supplementation affects over 38% of athletes who supplement protein periodically and 6.5% constantly [12].

Due to the wide spectrum of health benefits, bee pollen is considered a “superfood”. The main hypothesis of the study was whether bee pollen is as safe as whey protein concentrate supplementation. In our research, we wanted to see if bee pollen could replace whey protein as a dietary supplement for athletes and to compare the safety and overall influence of whey protein and bee pollen on rat liver. Both are freely and easily available dietary supplements. Most of the published studies on the impact of bee pollen focus on its antioxidant and anti-inflammatory properties [1,13,14,15,16,17,18]. None of them examine the impact on the liver; thus, in the current study, we want to fill-in this gap.

**Aim of the study**: The objective of the study is to assess the impact of dietary supplementation of bee pollen and whey protein on liver histological structure and expression of interleukin 12, smooth muscle actin and nitric oxide synthases of Wistar rat, reared without or under controlled systematic physical exercise conditions and to check whether bee pollen or whey protein affect oxidative stress levels.

## 2. Material and Methods

The study included 30 eight-week-old rats. The rats averaged 330 g at the beginning of the study and about 400 g at the end. Thirty male Wistar rats were divided into 6 equal groups—3 running: Con-Run (2), WP-Run (5), BP-Run (6) and 3 non-running: Con-Sed (1), WP-Sed (3), BP-Sed (4)—among them there were two control groups (Con-), two supplemented with bee pollen (BP-) and two receiving whey proteins (WP-) (Table 1).

In the present study, multi-flower bee pollen from the vicinity of Lublin (Poland), collected in the period July–August and an enriched whey protein concentrate (Olimp Laboratories Sp. z o.o., Dębica, Poland) were used. A total of 100 g of this supplement contained 77 g protein, 6 g carbohydrates, 7 g lipids, and 10 g water. Total methionine content was 0.8 g per 100 g of whey protein. On average, 100 g of bee pollen contains approximately 23 g of protein, 31 g of carbohydrates, 5 g of lipids, a total of about 0.8 g of vitamins (A, E, D, B1, B2, B3, B5, B6, B7, C), and approximately 40 g water [19]. Approximately 13.4 g and 12 g of bee pollen were eaten per day by each rat of the BP-Run and BP-Sed group, respectively. Each rat from the WP-Sed and WP-Run group ate, on average, 5.2 g of whey protein per day. Permanent access to water, standard feed, whey protein and bee pollen was provided. Food and supplements were weighted twice a day to determine how much was eaten each day per rat. During the 8-week laboratory phase, rats were weighted 15 times. All rats in running groups were running 5 times a week, 5 min each time, with a mean velocity of 6 km/h; no electrical shock was needed. The study protocol was as described previously, including detailed consumption data [20]. Animals were cared for in accordance with the Guide for the Care and Use of Laboratory Animals [21]. After the laboratory phase, all animals were decapitated and their livers were collected, formalin-fixed and paraffin-embedded. Five micrometer-thick slides were prepared and used for classical histological stainings incl. hematoxylin and eosin, Masson’s trichrome and periodic acid Schiff (PAS).

Immunochemistry (IHC) was performed as previously described by Zarobkiewicz et al. [20]. The following antibodies were used: anti-α-Smooth Muscle Actin (α-SMA, Elabscience, Houston, TX, USA, E-AB-33323, dilution 1:200), anti-interleukin 12β (IL-12β, Biorbyt, orb184406, dilution 1:100), anti-nitric oxide synthase 1 (NOS1/nNOS, Elabscience, Houston, TX, USA, dilution 1:100), and anti-nitric oxide synthase 3 (NOS3/eNOS, Elabscience, Houston, TX, USA, dilution 1:100) and anti-transforming growth factor β (TGF–β, St John’s Laboratory, dilution 1:200). Proteinase K (Novocastra, Leica Biosystems Newcastle Ltd., England, UK) was used to expose antigenic sites and the incubation lasted for 5 min. Endogenous peroxidase activity was blocked by 0.3% solution of perhydrol in methanol. Nonspecific binding was prevented by the addition of normal serum and incubation for 30 min. The material was incubated with primary antibody for 60 min and afterwards for another 30 min with horseradish-peroxidase-conjugated secondary antibody. Diaminobenzidine was used to visualize the reaction and hematoxylin was used to counterstain nuclei.

Slides were evaluated under a light microscope. Sections were digitally photographed (Olympus BX-42 and CellSens Software). Digital images (magnification of ×400) were analyzed using image analysis software: ImageJ was used to measure optical density and immunohistochemistry profile coverage as previously described [22]. In order to accurately compare the results of IHC for α-SMA, IL-12 β, nNOS, and eNOS, a statistical analysis based on IHC optical density score was performed, as previously described [23,24,25]. Statistica 13 was used for statistical analysis. Data distribution was assessed with the Shapiro–Wilk test. Kruskal–Wallis test with Bonferroni correction was used to calculate statistical significance between each group in multiple-comparison tests [26]. The level of significance was set as *p* < 0.05.

The study protocol was approved by the Bioethical Committee at the Medical University of Lublin (No. 24/2015).

## 3. Results

### 3.1. Morphological Changes

Microscopic evaluation of hematoxylin and eosin samples revealed no evident changes in the architecture of rat liver. Examination of Masson’s trichrome staining revealed areas of no specific fibrosis. Small differences in hepatocyte nuclei size and central vein diameters between some groups were also visible. Those changes were precisely measured in ImageJ and presented in Table 2.

#### 3.1.1. Dimensional Analysis of Histologically Significant Structures: Hepatocyte Nuclei and Central Veins

A significantly higher diameter of hepatocyte nuclei was observed in both experimental non-running groups compared to the non-running control group. Differences in running groups, albeit noticeable, were not significant (Table 2).

A greater diameter of central veins in all experimental groups compared to the respective control group were observed. The BP-Sed group had a larger diameter of central veins compared to the respective running group. An inverse relationship was noted in the whey protein supplemented groups. A significantly greater diameter was observed in all experimental groups compared to the Con-Sed group (Table 2).

#### 3.1.2. Collagen Deposition and Glycogen Content in Livers Analysis

To further evaluate the effect of supplementation on the metabolism and physiological architecture of the liver, Masson’s trichrome and PAS staining were performed (Figure 1 and Figure 2). The evaluation of Masson’s trichrome stained slides revealed no evident fibrosis. In all the experimental groups, OD values of Masson’s trichrome staining were calculated, and they were lower compared to the respective control group. A statistically significant decrease in collagen deposition in microscopic appearance was observed in the WP-Run group compared to the Con-Run group. In PAS staining, a statistically significant decrease in calculated OD value of glycogen content was observed in all experimental groups compared to the Con-Sed group (Table 3).

### 3.2. Immunohistochemistry Evaluation

#### 3.2.1. IL-12 Expression

To assess the inflammatory response, immuno-staining against interleukin 12 was performed. Significantly lower expression of IL-12 was observed in the running control group compared to the non-running control group; no other significant differences were noted (Table 3, Figure 3).

#### 3.2.2. eNOS and nNOS Expression

To assess the endothelial function, immuno-stainings against nNOS and eNOS were performed (Figure 4 and Figure 5, respectively). nNOS expression was lower in all experimental groups and Con-Run compared to the Con-Sed group. A significantly greater eNOS optical density score was noted in the running control group than the non-running. eNOS expression was the highest in the Con-Run group. The BP-Sed group had a significantly lower expression of eNOS than the Con-Sed group. Bee pollen supplemented groups had lower eNOS expression than the corresponding controls. Both eNOS and nNOS expression were noticeably lower in the bee pollen supplemented groups compared to the whey protein supplemented groups (Table 3).

#### 3.2.3. α-SMA Expression

Microscopic appearance revealed no significant difference in α-SMA expression between all groups (Figure 6). OD of α-SMA showed no significant changes in intensity of staining in addition to increased expression in the BP-Run group, compared to the Con-Run group (Table 3).

## 4. Discussion

The size of the nucleus is an indicator of the functional activity of the cell [27]. Therefore, the observed increase in the size of the nucleus suggests that these cells are probably hyperactive. Bee pollen supplementation in mice may lead to a decrease in total cholesterol (TC), triacylglycerols (TAG) and low-density lipoprotein (LDL cholesterol) [28]. Possible protective mechanisms might be induced by bee pollen or whey protein supplementation, influencing hepatic metabolism and partially protecting against hepatocyte degeneration. Similarly, its hepatoprotective role may manifest itself in low hepatic glycogen distribution.

The portal venous system plays a crucial role in the regulation of hepatic sinusoidal blood flow. The increasing diameter of the hepatic central vein (known also as terminal portal venule) is one of the parameters that reveals the tendency to portal hypertension [29]. Wider terminal portal venules seem to be related to lower blood pressure in the portal venous system [30]. Thus, it seems that bee pollen may lower portal venous blood pressure.

The evaluation of Masson’s trichrome and PAS stainings suggest that bee pollen and whey protein consumption may have an influence on hepatic fibrosis and decrease glycogen content. The authors in [16,18] maintain that whey protein may have a hepatoprotective effect. However, lack of evident fibrosis and a decreased level of collagen deposition in the rats’ livers may be the result of the antioxidant and anti-inflammatory properties of bee pollen and whey protein. This may be an interesting topic for further studies with experimentally-induced liver fibrosis.

Immunohistostaining against α-SMA did not reveal any significant changes. α-SMA expression correlates with activation of stellate cells and their differentiation to myofibroblast-like cells, which are responsible for liver fibrogenesis, which may lead to fibrosis and, consequently, cirrhosis [31]. In our research, a low level of hepatic fibrosis was observed and, similarly, α-SMA expression was low.

IL-12 is one of the most important proinflammatory cytokines presented with the initiation of immune response, determining Th1 and Th2 lymphocytes differentiation. Hence, it can be considered as a marker of initiation of inflammatory processes which can lead to liver fibrosis [32,33]. IL-12 is mainly produced by antigen-presenting cells as a result of interferon gamma (IFN-*γ*) stimulation [34].

It is presented with the initiation of immune response and has a central role in coordinating innate and adaptive immunity [35]. In our research, the non-running groups had statistically higher IL-12 expression than the running control group. This may be the result of beneficial properties of physical exercises. Myokines are a type of cytokine-produced and released by muscle cells during contractions. They may cause a decrease in IL-12 expression and a subsequent decline in inflammation [36]. Non-running experimental groups that received both bee pollen and whey protein had lower IL-12 expression than the control non-running group. However, it was not statistically significant. In the case of IL-12 expression, anti-inflammatory activity of bee pollen and whey protein is uncertain [18,28]. Experimental running-groups had lower IL-12 expression than experimental non-running groups, but this was also not statistically significant.

Nitric oxide (NO) is an important determinant of blood vessel tone and hepatic blood supply. NOS3 (eNOS) is mainly expressed in liver sinusoidal endothelial cells (LSECs) and endothelial cells of the hepatic artery, portal vein and central vein. eNOS is constitutively expressed and produces small amounts of NO in response to stimuli such as flow shear stress and vascular endothelial growth factor (VEGF). eNOS-derived NO maintains liver homeostasis and inhibits perfusion disorders, which may lead to ischemia and, subsequently, fibrogenesis in the liver [14,37]. In the experimental model of induced hepatic fibrosis, eNOS expression decline was observed. Thus, reduced expression of eNOS in chronic liver disease can reduce hepatic perfusion and accelerate fibrosis [38]. In our research, the non-running group which received bee pollen (BP-Sed) had significantly lower expression of eNOS in comparison to the non-running control group. Moreover, the running bee pollen group (BP-Run) had significantly lower expression of eNOS in comparison to the running control group. Bee pollen may have a negative impact on eNOS expression and hepatic perfusion. In our research, eNOS expression was statistically higher in the running control group than in the non-running control group, suggesting that running may have a positive impact on blood flow in the liver.

NOS1 is abundantly expressed in neurons and associated with the control of neuronal functions. However, it is also expressed in non-neuronal cells, such as endothelium and smooth muscle cells in blood vessels [39]. eNOS is considered as a main isoform involved in the regulation of vessel function. Nevertheless, studies have shown that nNOS present in vascular endothelium takes part in the regulation of cardiovascular functions [40]. In our research, a statistically significant difference was observed between the non-running control group and other groups. Likely, nNOS may compensate for the eNOS deficiency [15]. Thus, nNOS expression is higher in the non-running control group. Whey protein seems to have no impact on eNOS and nNOS expressions.

Bee pollen contains a relatively high number of various polyphenols and flavonoids, which are known to have significant antioxidant activity [3,5,6,41]. Polyphenols and flavonoids determine this property by reducing the level of reactive oxygen species (ROS) [38]. Lack of balance between the production of ROS and the antioxidants is associated with a variety of chronic health problems, including cardiovascular diseases, diabetes and degenerative hepatic diseases [14,38]. Many authors have demonstrated that polyphenols may help prevent alcoholic liver injury via antioxidation [13,15,16,28,42]. Thus, antioxidant properties of bee pollen could contribute to reduced production of ROS and to protection against oxidative stress injury. Polyphenols and flavonoids in bee pollen can contribute to the NO expression in contrast to the whey protein, where this bioactivity was not expectable. However, this relationship was not observed in our study.

## 5. Conclusions

The study, as designed, evaluated and compared the effects of dietary supplementation with bee pollen or whey protein on liver histological structure and expression of interleukin 12, smooth muscle actin, and nitric oxide synthase in rats supplemented with bee pollen and whey protein from the Wistar rat, reared without or with controlled, systematic exercise.

Bee pollen appears to be an interesting alternative to whey protein because of the increased cellular activity mediated by larger nuclei size and reduced glycogen deposition. The larger size of cell nuclei in bee pollen-supplemented groups, compared to whey protein-supplemented groups, indicates greater cellular activity of hepatocytes in these groups.

In rats reared under running conditions, the diameters of central veins in the liver reached smaller sizes when supplemented with pollen, than when supplemented with whey protein. Such an effect was not observed in rats reared under non-running conditions.

Bee pollen has less or no effect on liver fibrosis than whey protein.

The antioxidant effect of BP conditioned on eNOS expression is worse than WP. No such effect was documented for nNOS expression.

The hepatoprotective effect manifested by reduced SMA and IL-12 expression is better seen in WP-supplemented groups in rats under running conditions and in BP-supplemented groups in non-running rats.

Thus, it can be concluded that the advantages of using bee pollen as an alternative to whey protein are questionable and require deeper and more methodologically refined research.

## Figures and Tables

**Figure 1 foods-11-01131-f001:**
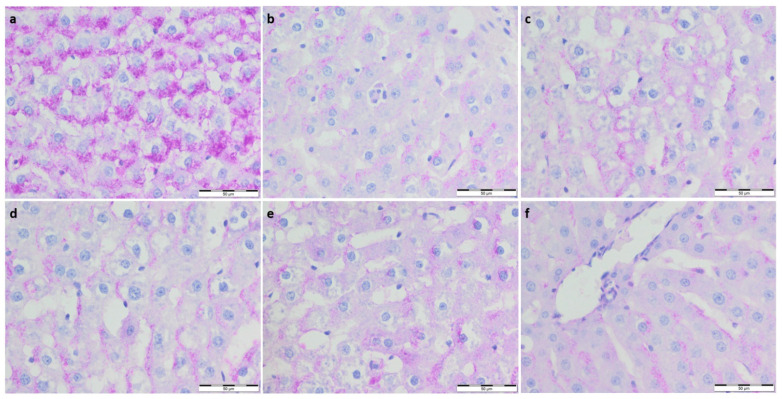
(**a**) Con-Sed, (**b**) Con-Run, (**c**) WP-Sed, (**d**) BP-Sed, (**e**) WP-Run and (**f**) BP-Run; BP-Run; Periodic acid Schiff; Glycogen was visualized using the periodic acid-Schiff reagent; Statistical decrease in glycogen deposition is visible in all experimental group compared to the Con-Sed group.

**Figure 2 foods-11-01131-f002:**
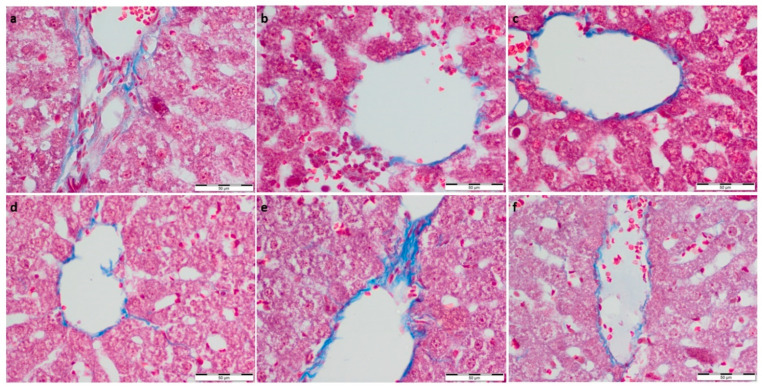
(**a**) Con-Sed, (**b**) Con-Run, (**c**) WP-Sed, (**d**) BP-Sed, (**e**) WP-Run and (**f**) BP-Run; Masson’s Trichrome Stain was intended for use in the histological visualization of collagenous connective tissue fibers in tissue sections; Statistical increase in collagen deposition is visible in WP-Run group compared to the Con-Run group.

**Figure 3 foods-11-01131-f003:**
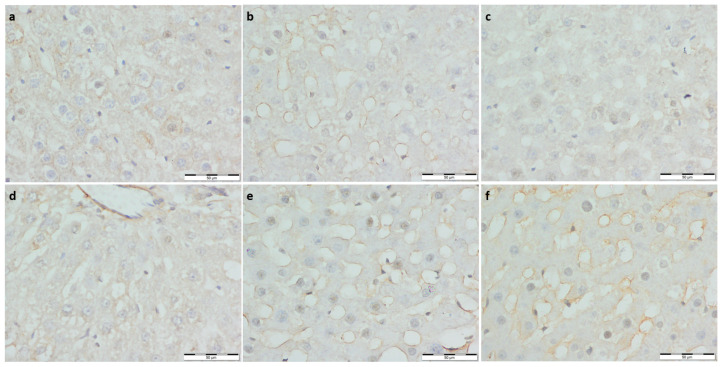
(**a**) Con-Sed, (**b**) Con-Run, (**c**) WP-Sed, (**d**) BP-Sed, (**e**) WP-Run and (**f**) BP-Run; immuno-stainings against Il-12; Significantly lower expression of IL-12 was observed in the running control group compared to the non-running control group, no other significant differences were noted.

**Figure 4 foods-11-01131-f004:**
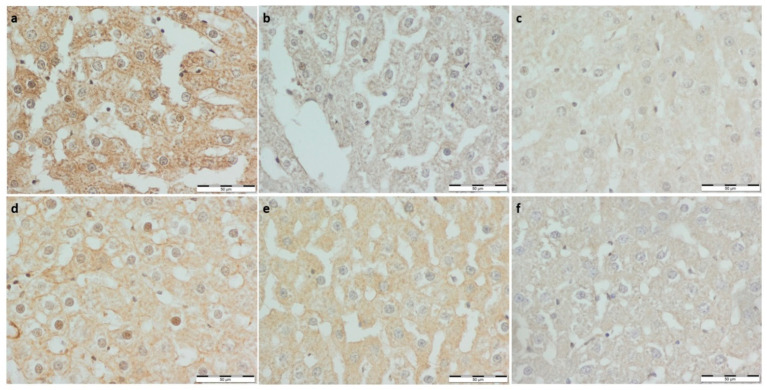
(**a**) Con-Sed, (**b**) Con-Run, (**c**) WP-Sed, (**d**) BP-Sed, (**e**) WP-Run and (**f**) BP-Run; immuno-stainings against eNOS (NOS3); Significantly greater eNOS optical density score was noted in the running control group than non-running. eNOS expression was the highest in the Con-Run group. The BP-Sed group had significantly lower expression of eNOS than the Con-Sed group. Bee pollen supplemented groups had lower eNOS expression than corresponding controls.

**Figure 5 foods-11-01131-f005:**
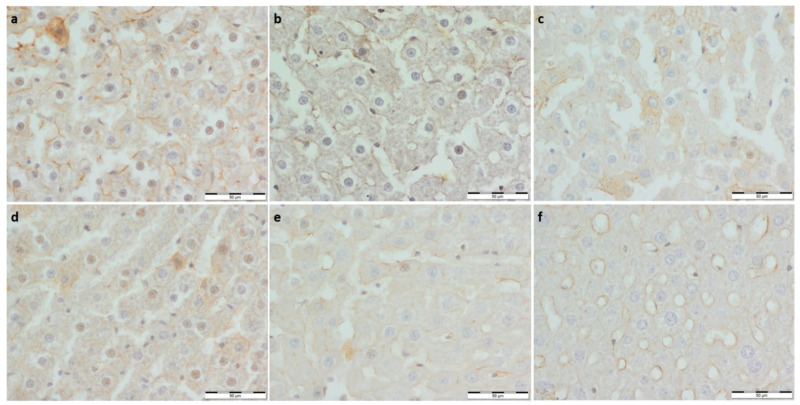
(**a**) Con-Sed, (**b**) Con-Run, (**c**) WP-Sed, (**d**) BP-Sed, (**e**) WP-Run and (**f**) BP-Run; immuno-stainings against nNOS (NOS1); nNOS expression was lower in all experimental groups and Con-Run comparatively to the Con-Sed group.

**Figure 6 foods-11-01131-f006:**
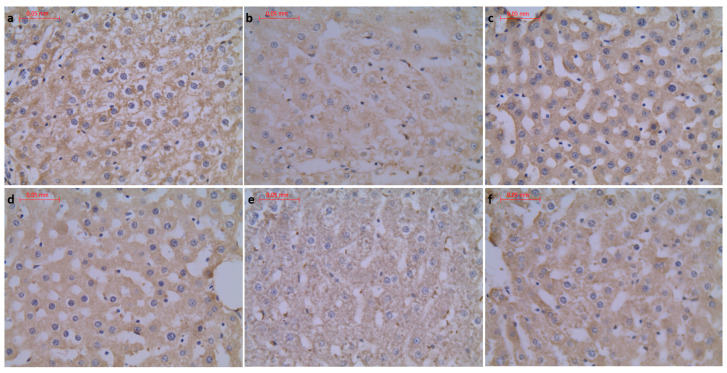
(**a**) Con-Sed, (**b**) Con-Run, (**c**) WP-Sed, (**d**) BP-Sed, (**e**) WP-Run and (**f**) BP-Run; immuno-stainings against α-SMA; OD analysis of α-SMA showed increased expression in WP-Run group compared to the Con-Run group.

**Table 1 foods-11-01131-t001:** Scheme of the study.

Group	Running	Supplementation	Group	Running	Supplementation
1 Con-Sed	No	No	2 Con-Run	Yes	No
3 WP-Sed	No	Whey protein	5 WP-Run	Yes	Whey protein
4 BP-Sed	No	Bee pollen	6 BP-Run	Yes	Bee pollen

**Table 2 foods-11-01131-t002:** The evaluation of nucleus and central vein diameter by groups.

	Con-Sed (1)	Con-Run (2)	WP-Sed (3)	BP-Sed (4)	WP-Run (5)	BP-Run (6)	^a^*p* < 0.05
Nucleus diameter [μm]	87.024 ± 12.85	101.1607 ± 12.03	95.9419 ± 123.47	104.7769 ± 13.97	100.4261 ± 14.37	101.3001 ± 16.69	1 vs. 5; 1 vs. 6
Central vein diameter [μm]	726.6297 ± 448.61	678.7059 ± 379.87	740.9134 ± 395.66	988.3653 ± 617.5406	845.2322 ± 522.1461	806.0831 ± 487.2394	1 vs. 3, 4, 5, 6; 2 vs. 3, 4; 3 vs. 4, 5, 6; 4 vs. 5, 6; 5 vs. 6

Note: ^a^ *p* values presented are the total *p* value (Kruskal–Wallis test); those for multiple comparisons were greater than 0.05.

**Table 3 foods-11-01131-t003:** The evaluation of optical density by groups in each of immunohistochemistry and Periodic Acid Schiff and Masson’s staining.

Optical Density (OD)	Con-Sed (1)	Con-Run (2)	WP-Sed (3)	BP-Sed (4)	WP-Run (5)	BP-Run (6)	^d^*p* < 0.05
SMA ^a^	1.235 ± 0.093	1.150 ± 0.079	1.253 ± 0.165	1.215 ± 0.182	1.239 ± 0.112	1.389 ± 0.274	2 vs. 6
Il-12 ^a^	1.493 ± 0.315	1.2 ± 0.27	1.376 ± 0.217	1.334 ± 0.190	1.195 ± 0.237	1.250 ± 0.262	1 vs. 2 1 vs. 5
nNOS ^a^	1.699 ± 0.133	1.512 ± 0.289	1.365 ± 0.46	1.246 ± 0.250	1.338 ± 0.321	1.26 ± 0.286	1 vs. 2, 3, 4, 5, 6
eNOS ^a^	1.113 ± 0.186	1.301 ± 0.325	1.174 ± 0.258	1.099 ± 0.0388	1.232 ± 0.318	1.109 ± 0.177	1 vs. 2, 4 3 vs. 2; 2 vs. 6
PAS ^b^	0.013 ± 0.048	0.011 ± 0.044	0.002 ± 0.001	0.002 ± 0.001	0.004 ± 0.002	0.003 ± 0.001	1 vs. 2, 3, 4, 5; 3 vs. 4, 5; 4 vs. 5; 5 vs. 6;
Masson ^c^	0.186 ± 0.087	0.227 ± 0.205	0.129 ± 0.04	0.193 ± 0.136	0.102 ± 0.021	0.146 ± 0.191	1 vs. 5; 2 vs. 5; 4 vs. 5

Note: ^a^ Optical density here is a measurement of the intensity of expression of the reaction; the higher the value, the more intense the expression. ^b^ Optical density here is a measurement of the intensity of hepatic glycogen deposition: the higher the value, the more intense the distribution deposition. ^c^ Optical density here is a measurement of the intensity of fibrosis; the higher the value, the more intense the fibrosis. ^d^ *p* values presented are the total *p* value (Kruskal–Wallis test); those for multiple comparisons were greater than 0.05.

## Data Availability

Data is contained within the article.

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
