# Peer review of "Changes in Histological Structure, Interleukin 12, Smooth Muscle Actin and Nitric Oxide Synthase 1. and 3. Expression in the Liver of Running and Non-Running Wistar Rats Supplemented with Bee Pollen or Whey Protein"

_foods, 2022, doi:10.3390/foods11081131_

Round 1

Reviewer 1 Report

Dear Authors

In your work “Changes in histological structure, interleukin 12, smooth muscle actin and nitric oxide synthase 1. and 3. expression in the 3 liver of Wistar rats supplemented with bee pollen or whey protein”, a research was made using two different matrices, which make the evaluation of the results confusing .

Despite that, is crucial to used no-process products as dietary supplements which were validated for that purpose.

For future investigation, in my point of view, the selection of the samples should have a better understanding for the reader. My advice is, that may be a small sentence explaining this choice could help in the present manuscript.

I include a few considerations directly in the pdf of the manuscript and the authors should analyze the pertinence of them and act as it is the best for the understanding of the work.Best regards

Dear Authors

In your work “Changes in histological structure, interleukin 12, smooth muscle actin and nitric oxide synthase 1. and 3. expression in the 3 liver of Wistar rats supplemented with bee pollen or whey protein”, a research was made using two different matrices, which make the evaluation of the results confusing .

Despite that, is crucial to used no-process products as dietary supplements which were validated for that purpose.

For future investigation, in my point of view, the selection of the samples should have a better understanding for the reader. My advice is, that may be a small sentence explaining this choice could help in the present manuscript.

I include a few considerations directly in the pdf of the manuscript and the authors should analyze the pertinence of them and act as it is the best for the understanding of the work.

Best regards

Author Response

Dear Reviewer,

Thank you very much for your review. We have made every effort to best present the results of the experiment conducted. We agree that the changes and explanations you suggested better compare the two types of supplementation. Changes related to figures will be applied by the editor, because I had a problem with putting them in the work in the original size. We have made changes to the introduction and purpose of the experiment and made thorough revisions to the conclusions. The conclusions have become clearer. We hope that the revised article will now be more understandable. 

Regards, 

Piotr Jarosz

Reviewer 2 Report

Review report:

Manuscript ID: foods-1646202

Title of the manuscript: Changes in histological structure, interleukin 12, smooth muscle actin and nitric oxide synthase 1. and 3. expression in the liver of Wistar rats supplemented with bee pollen or whey protein.

The authors presented the results of histological structure and changes in IL-12, SMA, NOS1 and NOS3 in trained and un-trained rats exposed to diet supplemented with  bee pollen or whey protein. The results of the study are interesting and are in the scope of Foods journal, however the current version of the text contains many shortcomings.

My decision: major revision

Detailed comments and suggestions:

  1. The manuscript needs a scientific proofreading;
  2. The Abstracts needs to be re-written. Some issues are described in details (e.g. histological staining and assessment, but the background and the aim of the study are presented too general.
  3. The aim of the study is too general and is inadequate to the presented results. It is not clear from the purpose of the study, why the animals were trained?
  4. Material and methods; The description of the animal experiment requires an additional information, such as: age of rat at the beginning of experiment, animal husbandry data, number of rats per group, composition of experimental diets.
    What was the level of bee pollen and whey protein in the experimental diets?
    Were the animals subjected to the procedure reducing the stress before decapitation or were they subjected to anesthesia?
  5. Results
    - in my opinion, the statistical analysis should reflect the assessment of the influence of two main experimental factors that were investigated (i.e.: dietary intervention and training). The method of analyzing the results presented in the paper makes it impossible to assess these factors.
    - the results of liver weights are highly recommended, as well as macroscopic analysis, food intake;

- Why the titles of subsections refers to obtained results? (e.g.: 3.1. No distinct morphological changes in any group; 3.1.1. Higher hepatocyte nuclei diameter in experimental groups and higher diameter of central veins in non-running groups …)

  1. Some parts of Discussion are speculative and do not refer to obtained results (e.g. lines 186-187)
  2. Conclusions need correction, because this is repetition of results or discussion.

Author Response

Dear Reviewer,

Thank you very much for reviewing the article. All the comments were extremely pertinent. I wanted to address each of them as thoroughly as possible.

In accordance with your suggestion, the article has undergone scientific correction. The abstract was rewritten. The research objective and methodology of the experiment were detailed.

Titles of paragraphs in the results section were changed. Due to the limitations of the way the experiment was planned and the limitations of the collected results, we have made the best possible objective analysis of the results, in the form of tables, allowing wide interpretation. The text focuses on statistically significant results. With the suggested changes, the rephrasing and presentation of the conclusions allowed a better interpretation of the results in the experiment. 

Indeed, the discussion contains sentences that are speculative in nature; however, there are not many studies on similar topics. Also, there are no studies that can confirm the results we obtained, thus, taking care of their best reliability, we have raised in the discussion some problems that we believe require further research and interest from the scientific world.

Regards, 

Piotr Jarosz 

Round 2

Reviewer 2 Report

The Authors properly addressed my comments and suggestions. I recommend the text for publication.